# Deep ensemble model for segmenting microscopy images in the presence of limited labeled data

**Jan Mikolaj Kaminski** [1]                                    JAN.KAMINSKI@SUND.KU.DK

**Ilary Allodi** [1]                                            IALLODI@SUND.KU.DK

**Roser Montañana-Rosell** [1]                                  ROSER.ROSELL@SUND.KU.DK

**Raghavendra Selvan** [1,2]                                    RAGHAV@DI.KU.DK

**Ole Kiehn** [1]                                               OLE.KIEHN@SUND.KU.DK

[1]*Department of Neuroscience & *[2]*Department of Computer Science, Univ. of Copenhagen, Denmark*

**Editors:** Under Review for MIDL 2021

## Abstract

Obtaining large amounts of high quality labeled microscopy data is expensive and time-consuming. To overcome this issue, we propose a deep ensemble model which aims to utilise limited labeled training data. We train multiple identical Convolutional Neural Network (CNN) segmentation models on training data that is partitioned into folds in two steps. First, the data is split based on sample diversity or expert knowledge reflecting the possible *modes* of the underlying data distribution. In the second step, these partitions are split into random folds like in a cross-validation setting. Segmentation models based on the U-net architecture are trained on each of these resulting folds yielding the candidate models for our deep ensemble model which are aggregated to obtain the final prediction. The proposed deep ensemble model is compared to relevant baselines, in their ability to segment interneurons in microscopic images of mice spinal cord, showing improved performance on an independent test set.

**Keywords:** Microscopy image, Segmentation, Ensemble model, Limited labeled data

## 1. Introduction

One of the fundamental assumptions in supervised machine learning is that the training data are sampled in an independent and identically distributed (i.i.d) manner, so as to approximate the underlying data distribution from samples (Abu-Mostafa et al., 2012). This might be true asymptotically when the number of data points increase but is *mostly* violated when working with smaller datasets. Further, when the data distribution has multiple modes either due to class imbalance or due to annotator variations, the task of supervised machine learning becomes ill-posed in the presence of limited training data.

In this work, we investigate the possibility of constructing a deep ensemble model using identical ensemble members that are trained to capture local modes of the training data, when i.i.d assumptions are not met. We apply the proposed method to segment interneurons from high resolution microscopy images in the presence of around 10% weakly annotated data. We estimate the segmentation performance on an independent test set with dense annotations, and compare its performance to relevant baseline methods.

## 2. Methods

The primary difference in the proposed ensemble method compared to existing methods (Lakshminarayanan et al., 2017) is in using identical models trained on non i.i.d training data; without constructing ensembles of different model types. To this effect, the training data is first partitioned into different folds in a two-step process. Details on partitioning of data and the construction of the Deep Ensemble model are presented next.

**Training data partitioning:** Consider a labelled training set: $\mathcal{D} = (\mathbf{X}, \mathbf{Y})$ with $N$ input images $\mathbf{X} = \{\mathbf{x}_i\}_{i=1}^N \in \mathbb{R}^{H \times W}$ and segmentation masks, $\mathbf{Y} = \{\mathbf{y}_i\}_{i=1}^N \in [0,1]^{H \times W}$.

In the first step of data partitioning, the dataset is split into $M$ partitions $\mathcal{D} = \cup_{m=1}^{M+1} \{\mathcal{D}^m\}$ where $\mathcal{D}^m = (\mathbf{X}^m, \mathbf{Y}^m)$. Each partition, $\mathcal{D}^m$, is assumed to correspond to a mode of the underlying multi-modal data distribution. Several criteria can be used for creating these partitions. For instance, splits can be made based on the expert knowledge of annotators in order to capture large variations within the training data. Another option, when feasible, is to treat different raters as a unique mode. This meta knowledge derived from the expert interactions can translate into meaningful modes of the data distribution.

The complete training dataset, $\mathcal{D}$, is also treated as a mode, resulting in a total of $M+1$ partitions (Figure 1-Column 1). Each of these $M + 1$ mode-level datasets, $\mathcal{D}^m$, are further randomly split into $K$-folds: $\mathcal{D}^m = \cup_{k=1}^K \{\mathcal{D}_k^m\}$ (Figure 1-Column 2).
**Deep Ensemble Model:** Performing $K$-fold cross-validation within each mode-level dataset, $\mathcal{D}^m$, yields $K$ converged models. As a result, for the $M + 1$ mode-wise datasets in the full pipeline, there are a total of $(M + 1) \cdot K$ segmentation models trained.

A segmentation network validated on the data split $\mathcal{D}_k^m$ is given as, $f_k^{(m)}(\cdot) : \mathbf{x} \mapsto \mathbf{y}$. The index $m$ is the mode-level and $k$ is the fold used for the model selection, when trained on the remaining $K - 1$ folds. Predicted segmentations at each mode are combined to obtain the first level ensemble based on cross-validation:

$$\mathbf{y}^{(m)} = \frac{1}{K} \sum_{k=1}^K \mathbb{I}[f_k^{(m)}(\mathbf{x}) > 0.5] \tag{1}$$

with $K$ non-zero levels.[1]

The final prediction from the proposed deep ensemble model is constructed by combining predictions from all the mode-level predictions, with $(M + 1) \cdot K$ discrete prediction levels:

$$\mathbf{y} = \frac{1}{(M + 1)} \sum_{m=1}^{M+1} \mathbf{y}^{(m)}. \tag{2}$$

## 3. Experiments and Results
**Data:** The objective of this work was to segment interneurons from the mice spinal cord in grey-scale microscopy images. Slice level images were acquired at two different time points: 1st- (P1) and 28th- (P28) postnatal days at resolutions 8320x8568 and 5262x6237, respectively. Patches of size 512x512 with 32px overlap were extracted from the slices resulting in 529 patches. Of these, weak annotations marking interneurons for 52 patches for P1 and 20 for P28 were provided, which was used for training. We treat patches from P1 and P28 images as the two mode-levels in this work i.e $M = 2$. Due to the difference in number of patches per mode, we use the following K-fold structure for P1, P28 and P1+P28: $K = [4, 5, 6]$, respectively. Thus, we obtain $(M + 1) \cdot K = 15$ data splits in total. Four patches from P1 and P28 were densely annotated which was used for test purposes.

For each of the cross validation settings within modes, we trained a standard U-net (Ronneberger et al., 2015) with 32 initial channels. Models were trained for a maximum of 300 epochs and optimized using Adam optimizer with a learning rate of $10^{-4}$. Early stopping was implemented to prevent overfitting with a patience of 30 epochs. The weights were

---

1. $\mathbb{I}[\cdot] : y \mapsto \{0, 1\}$ is the indicator function.

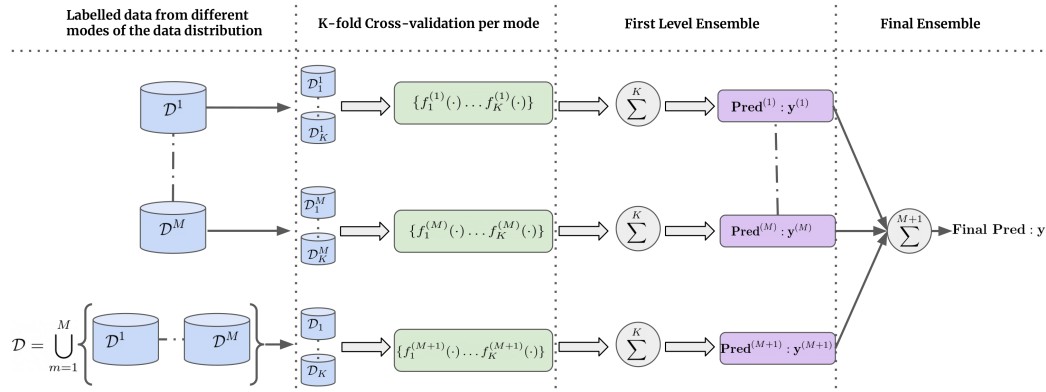

Figure 1: Pipeline for the proposed Deep Ensemble model. The training dataset is split into $M$ partitions; an additional combined dataset is obtained comprising all $M$ partitions (last row). Each of these $M + 1$ partitions are trained in a K-fold cross validation setting. Predictions from each cross-validation are aggregated to obtain the first level ensemble prediction. Finally, the $M + 1$ predictions are combined to output the final model prediction.

optimised based on Dice loss and F1 score was used to compute the segmentation accuracy. The proposed ensemble model was compared to a standard U-net and Seeded Region Growing (SRG) method, which were trained on all training data $\mathcal{D}$. The baseline U-net was trained with 6-fold cross-validation and test set performance was estimated over all folds.

**Results and Discussions**: Table 1 shows the test set performance of the proposed ensemble method compared to SRG and baseline U-net. The difference in F1-score ($\approx 0.1$) clearly shows the strength of the proposed ensemble method.

The threshold for the proposed ensemble model that resulted in best binary segmentation was the lowest possible discrete level (1/15). This implies that all models contribute in improving the overall F1 score. In this regard, the proposed ensemble method with identical networks trained on different data distributions, is able to learn complementary information. We further hypothesize that the proposed ensemble model can be most beneficial in non-i.i.d settings; in i.i.d settings when all data folds are identically distributed, networks of identical architecture, would end up learning similar segmentation models. We conclude that the diversity of the proposed ensemble model is achieved by learning on different modes of the data distribution.

Table 1: Performance comparison on test set

| Method | F1 score |
|---|---|
| Ensemble model | **0.64 ± 0.10** |
| Standard U-net | 0.53 ± 0.02 |
| SRG | 0.59 ± 0.18 |

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
