# OpenReview forum: "Deep ensemble model for segmenting microscopy images in the presence of limited labeled data"
_MIDL.io/2021/Conference/Short — MIDL 2021 Poster_

### Official Review · Reviewer_Z9hM · 2021-04-30

**Confidence:** 4
**Final Rating:** 3

**Summary:**

The authors train multiple identical segmentation models on limited training data which is partitioned into folds in two steps. First, the data is split based on sample diversity or expert knowledge reflecting the underlying distribution. Second, these partitions are split into random folds to train multiple U-Nets. Results are aggregated to obtain the final prediction.

**Strengths:**

The authors show that cross-validation-based ensemble learning drastically improves segmentation results in a image segmentation task, especially reducing the false postives.
The partitioning and ensemble strategies sound reasonable.

**Weaknesses:**

There are a few weaknesses from a technical view, but not totally weaknesses as this is a short paper.
1. A simple comparison with the naive ensemble [1, 2] is missing, i.e., training multiple identical networks with the whole training set and perform major voting.
2. The authors use K=15 and M=2. It seems that it is very computationally expensive.
3. It was great to see F1 score improved which means the ensemble reduced false positives. What about the other metrics for example, Dice score?


References:
[1] Ensembles of Multiple Models and Architectures for Robust Brain Tumour Segmentation
[2] Fully convolutional network ensembles for white matter hyperintensities segmentation in MR images

**Deanonymize Review:**

no

**Detailed Comments:**

Mainly from the weakness part.

1. A simple comparison with the naive ensemble is [1,2] missing, i.e., training multiple identical networks with the whole training set and perform major voting.
2. The authors use K=15 and M=2. It seems that it is very computationally expensive.
3. It was great to see F1 score improved which means the ensemble reduced false positives. What about the other metrics for example, Dice score?

References:
[1] Ensembles of Multiple Models and Architectures for Robust Brain Tumour Segmentation
[2] Fully convolutional network ensembles for white matter hyperintensities segmentation in MR images

**Justification Of The Rating:**

1. The paper demonstrates that a cross-validation based ensemble strategy is effective in image segmentation.
2. A simple comparison with the naive ensemble is missing
3. Generally the paper is well written.

**Paper Type:**

validation/application paper

**Special Issue:**

no

---

### Official Review · Reviewer_YjKH · 2021-04-30

**Confidence:** 4
**Final Rating:** 3

**Summary:**

The paper presents a deep ensemble modeling strategy for effectively learning patterns from different modes of data arising due to annotator variations or class imbalance. The authors propose to train identical models on different modes of data and average their performance on the test dataset for final prediction. The strategy is evaluated on mice spinal cord microscopy images acquired at two different time points: 1st and 28th postnatal days (2 modes) and are giving 0.1 average F1 score gain compared to the traditional approach.

**Strengths:**

* The paper proposes an interesting method for incorporating information from different modes of data for model training. Data modalities like variation in annotations arising due to multiple data annotators or splits created based on domain expertise etc. are often observed in medical data. This approach can help incorporate this information for limited data training.


**Weaknesses:**

* For the Ensemble model, the authors have used all the data in their (M+1)st fold with six-fold cross-validation for test data prediction; the same set-up is used for Standard U-net training. However, the average gains seen in the Ensemble model is approximately 0.1 compared to the standard U-net model, indicating that gains are coming from U-net models trained on P1 and P28 splits. Hence it becomes important to report model performance when trained on only P1 or P28 split to understand where the gains in the performance are coming from and how much the variations in different modes of data impact model performance.
* The performance for the proposed strategy has been reported for a single small mice spinal microscopy dataset. It might not hold when expanded to other datasets or large mice spinal microscopy datasets. More evaluation is required for studying the potential of the proposed strategy on limited data problems.


**Deanonymize Review:**

no

**Detailed Comments:**

* It is often observed that ensembling multiple models or the same model with different seeds leads to performance gains. Hence, the gains observed in the paper are not surprising. However, the use of multiple models on different modes of data is new and can help draw interesting insights from the data.
* Validation of the proposed approach is limited in the paper, and further evaluation is required. The gains can be because of differences in P1 and P28 data distribution, which leads to a better model when learned independently. Hence, the model's performance on P1 and P28 should be reported in the validation table, and a high standard deviation in the performance should be investigated.

**Justification Of The Rating:**

The paper proposes a strategy for effectively using multiple modes of data for model development in the limited data problem. There are few limitations in the paper that can be improved, but overall the idea is interesting.  More experiments on different medical problems are required to understand the method's potential.

**Paper Type:**

methodological development

**Special Issue:**

no

---

### Meta-Review · Area_Chair_cJNe · 2021-05-07

**Recommendation:** Accept (Poster)
**Confidence:** 5

**Metareview:**

The reviewers agree that, despite some open questions, the paper is well-written and should be accepted.

---

### Decision · Program_Chairs · 2021-05-11

Accept (Poster)